# Apigenin and Luteolin Regulate Autophagy by Targeting NRH-Quinone Oxidoreductase 2 in Liver Cells

**DOI:** 10.3390/antiox10050776

**Published:** 2021-05-13

**Authors:** Elzbieta Janda, Concetta Martino, Concetta Riillo, Maddalena Parafati, Antonella Lascala, Vincenzo Mollace, Jean A. Boutin

**Affiliations:** 1Department of Health Sciences, Campus Germaneto, Magna Graecia University, 88100 Catanzaro, Italy; concettamartino83@gmail.com (C.M.); criillo@unicz.it (C.R.); mparafati@unicz.it (M.P.); anto.lascala@gmail.com (A.L.); mollace@unicz.it (V.M.); 2Interregional Research Center for Food Safety and Health, 88100 Catanzaro, Italy; 3PHARMADEV (Pharmacochimie et Biologie Pour le Développement), Faculté de Pharmacie, Université Toulouse 3 Paul Sabatier, 31062 Toulouse, France; ja.boutin.pro@gmail.com

**Keywords:** polyphenol, quinone reductase, menadione, autophagy, flavoenzyme, IC50, enzymatic activity, bergamot flavonoids

## Abstract

Dietary flavonoids stimulate autophagy and prevent liver dysfunction, but the upstream signaling pathways triggered by these compounds are not well understood. Certain polyphenols bind directly to NRH-quinone oxidoreductase 2 (NQO2) and inhibit its activity. NQO2 is highly expressed in the liver, where it participates in quinone metabolism, but recent evidence indicates that it may also play a role in the regulation of oxidative stress and autophagy. Here, we addressed a potential role of NQO2 in autophagy induction by flavonoids. The pro-autophagic activity of seven flavonoid aglycons correlated perfectly with their ability to inhibit NQO2 activity, and flavones such as apigenin and luteolin showed the strongest activity in all assays. The silencing of NQO2 strongly reduced flavone-induced autophagic flux, although it increased basal LC3-II levels in HepG2 cells. Both flavones induced AMP kinase (AMPK) activation, while its reduction by AMPK beta (PRKAB1) silencing inhibited flavone-induced autophagy. Interestingly, the depletion of NQO2 levels by siRNA increased the basal AMPK phosphorylation but abrogated its further increase by apigenin. Thus, NQO2 contributes to the negative regulation of AMPK activity and autophagy, while its targeting by flavones releases pro-autophagic signals. These findings imply that NQO2 works as a flavone receptor mediating autophagy and may contribute to other hepatic effects of flavonoids.

## 1. Introduction

Autophagy is an intracellular pathway that targets and delivers cellular components such as organelles and protein aggregates to lysosomes for degradation. Autophagy dysfunction has been identified as a key factor in the pathogenesis of several diseases, ranging from neurodegenerative disorders, such as Parkinson disease to metabolic disorders, such as nonalcoholic fatty liver disease (NAFLD) [1,2,3]. Indeed, many of the functions of autophagy are impaired in NAFLD, including its ability to regulate cellular insulin sensitivity, mediate hepatocyte resistance to harmful stimuli such as oxidants and cytokines, and metabolize cellular lipid stores by lipophagy [4,5,6]. Natural polyphenols and in particular some flavonoids from *Citrus* family fruits such as bergamot have been shown to stimulate autophagy in liver cells [7,8] and prevent NAFLD in animal models and clinical studies [9,10,11]. The pro-autophagic activity of these compounds has also been documented in several other cell and tissue types [12,13,14], and now, it is widely accepted that many flavonoids, alkaloids, and other polyphenols stimulate autophagy, besides their well-known antioxidant and anti-inflammatory effects [15,16,17].

Several molecular mechanisms have been proposed to explain the biological functions of flavonoids. According to the oldest, but highly controversial view, flavonoids would exert their effects by free-radical scavenging [16,18,19], which is proportional to the redox potential of these molecules, usually correlating with the number of hydroxyl groups [20,21,22]. However, the last two decades have provided more convincing explanations of antioxidant effects of flavonoids, which suggest that these compounds (i) interact with specific proteins central to signaling cascades and modulate the activity and/or expression of key antioxidant proteins; (ii) influence the epigenetic mechanisms of gene expression; and (iii) modulate the gut microbiota profile and metabolites. Thus, while the current view is that pleiotropic mechanisms contribute to the final beneficial effect of polyphenols [21,22], the role of specific flavonoid–protein interactions is increasingly recognized [23]. A well-recognized direct target of polyphenols are phosphodiesterases (PDEs). For example, it has been shown that resveratrol inhibits PDE1, PDE3, and PDE4 and thereby mediates the increase of cAMP levels and AMPK activation via the Epac and CamKKB pathway, leading to metabolic benefits such as the prevention of diet-induced obesity and diabetes. The same authors showed that resveratrol binds to PDE3 in two different conformations and thereby competes with cAMP for the access to the catalytic pocket of PDE3 [24,25].

PDEs are not unique direct targets of flavonoids, and computational studies predict over 3000 proteins that may bind different polyphenols [23], but only a few polyphenol–protein interactions were characterized in detail. Most of mechanistic studies address the effects of flavonoids on downstream signaling pathways, but they disregard direct molecular targets of these compounds. Therefore, little is known on which are biologically the relevant targets of polyphenols and how they sense and convert the interaction with polyphenols into pharmacological effects.

Among these targets, NQO2, also known as QR2 (quinone oxidoreductase 2), appears well characterized in a series of crystallographic studies, yet very few papers convincingly show that the physical interaction with flavonoids or other polyphenols causes a specific biological function [26]. NQO2 binds to and is inhibited by resveratrol by a direct physical interaction [27]. After this seminal work, further studies showed that NQO2 binds to other natural and synthetic stilbene, coumarin and flavonoid derivatives. Some of these compounds, such as quercetin, and diosmin were shown to potently inhibit the catalytic activity of NQO2 at sub-micromolar concentrations [28,29].

NQO2 is a cytoplasmic flavoenzyme catalyzing two-electron reductions of endo- and xenobiotic quinones. Similar to other Phase I and II drug metabolizing enzymes, it is regulated by Nrf2 and other transcription factors that bind to antioxidant response elements (ARE) present in the NQO2 promoter region [26,30]. The most peculiar feature of NQO2 is its inability to recognize NADH, but N-alkyl nicotinamide derivatives as its co-substrates, in contrast to its next of a kind, NADH: quinone oxidoreductase 1 (NQO1). NQO2 is highly expressed in liver and kidney, where it may play both a detoxifying as well as a toxifying function, depending on the substrate [26]. In this regard, NQO2 has been shown to mediate the toxic effect of certain quinones mainly in ortho conformation as well as other non-quinonic compounds, such as parkinsonian toxins, the antitumor drugs CB154 and mitomycin [26,31,32]. The biological roles of NQO2 may involve other processes beside the reduction of quinones and drug metabolism, such as the regulation of oxidative stress and cell death [32], protein stability [33], inflammation, and memory formation in the cortex [34,35], but some of these findings need further consolidation.

In addition, our previous work demonstrated that a synthetic and highly specific NQO2 inhibitor (NMDPEF/S29434) is a potent inducer of autophagy in astrocytes [36] and was able to induce autophagy in hepatocytes, and in both cases, this effect was shown to be NQO2-dependent [36,37]. Since different flavonoids have been shown to be natural inhibitors of NQO2 [28], we hypothesize that NQO2 may also play a role in flavonoid-induced autophagy. In the present work, we demonstrate that the pro-autophagic activity of different Citrus flavonoid aglycons correlates with their capacity to inhibit NQO2 oxidoreductase in liver cells, indicating apigenin and luteolin as the most active in both assays. Furthermore, we show that silencing of NQO2 in HepG2 cells reduces their responsiveness to both flavones in terms of autophagic flux. We further show a possible involvement of AMPK in this process. These data suggest for the first time that this direct target of flavonoids plays a role in flavone-induced autophagy, which is likely mediated by AMPK activation.

## 2. Materials and Methods

### 2.1. Cell Culture

HepG2 cells were a kind gift of Dr. Antonio Brunetti, University Magna Graecia, Catanzaro. HepG2 cells were cultured in 4.5% glucose DMEM (Carlo Erba srl, Milan, Italy, FA30WL0101500 supplemented with 10% Fetal Bovine Serum (FBS, Life Technologies (Tech.), Monza MB, Italy, 10500-064), 1:100 Penicillin–Streptomycin (Pen-Strep, Life Tech., 15070-063), and 1:100 L-Glutamine (PAA Laboratories GmbH, Austria, M00409-2722). For experiments, HepG2 cells were seeded 3 days before treatments at the density 36,000 × cm^−2^ (350,000 per 3.5 cm plates), and the last medium change was performed 24 h before the treatment with flavonoids.

### 2.2. Reagents

Flavonoid aglycons (except quercetin) were purchased from Extrasynthese (Genay Cedex, France) and were diluted in ethanol 100% to stock solutions 10 mg/mL, except for diosmetin that was diluted as 2 mg/mL stock due to its low solubility. Chloroquine (25 mM stock in PBS), palmitic acid (PA) (0.3 M stock in ethanol) and quercetin (25 mM stock in ethanol) were from (Merk, Sigma-Aldrich, Darmstadt, Germany). The compounds were kept in aliquots at −20 °C and thawed shortly before each treatment. The antibodies (Abs) used for Western blotting (WB) were polyclonal rabbit anti-LC3A/B (Abcam plc, Cambridge, UK, ab1280025), anti-p62 (MBL International, Woburn, MA, USA, PM045), anti-GAPDH, clone FL335 (Santa Cruz Biotech., Dallas, Texas, USA, sc-25778), anti-NQO2 (Proteintech, Manchester, UK), and monoclonal mouse anti-α-Tubulin (Merk, Sigma-Aldrich, T6074). Ab used to detect phosphorylated (ph)AMPK alpha 1 was anti-AMPK alpha1 (phospho-T183) + AMPK alpha2 (phospho-T172) antibody (Abcam, ab133448).

### 2.3. Generation and Flow Cytometry Analysis of GR-LC3-HepG2 Cells

HepG2 cells expressing DsRed-LC3-GFP (GR-LC3-HepG2) were generated as previously described [7,38]. Cells were seeded 3 days before treatments on 24-well plates (in triplicate) and then treated with PA or ethanol for 24 h and next day flavonoids or ethanol (vehicle) were added 6 h before harvesting (trypsinization), as previously described for flow cytometry analysis of viability [32]. After one wash in PBS, cells were pelleted and resuspended in 0.45 mL PBS containing 1% FBS and 0.1 mM EDTA (Merk, Sigma-Aldrich, E5134). Cells were acquired in 502 nm (FITC, green) and 556 nm (PE, red) channels by FACSCanto II (BD Biosciences, Erenbodgem, Belgium), and mean fluorescence intensity (MFI) was analyzed in a selected population of intermediate size singlets. The ATG index was calculated as the mean ratio between the PE-A MFI and FITC-A MFI values (red/green MFI ratio).

### 2.4. HepG2 Transfection and Silencing

HepG2 cells were transfected with a reverse transfection protocol described earlier [37] with several modifications. Non-silencing (control, All-stars, 1027280) were purchased from Qiagen (Milan, Italy); esiRNA for human NQO2 were from Merk, Sigma-Aldrich. Transfection was performed using Lipofectamine 2000 and Optimem (51985026) both from Life Tech., according to the manufacturer’s instructions. siRNA (100 nM) and lipofectamine (were diluted in Optimem (150 µL) and incubated in one well of the 12-well plate (15–20 min at room temperature, RT). The cells were trypsinized, suspended in standard medium without Pen-Strep, and reverse transfection was carried out by the addition of 600 µL of cell suspension (300,000 cells/well) to the preincubated transfection complexes and incubation for 18 h. After 18 h, the medium was changed (supplemented with Pen-Strep), and cells were incubated for the next 24 h before the desired treatment.

### 2.5. NQO2 Activity Measurements

HepG2 cells or mouse (BL6/129, 3 months old) liver protein extracts were performed in in 50 mM Tris–HCl, containing 1 mM n-octyl-d-glucopyranoside pH 8.5 buffer (QR buffer). For this purpose, 100 mg of frozen liver was homogenized in 1 mL of QR buffer with 30 strokes of tight glass douncer on ice, centrifuged once 5 min at 1000× *g*, and then 15 min at 14,000× *g*, at 4 °C. Alternatively, one confluent 10 cm plate of HepG2 cells was processed in 500 μL of QR buffer, as previously described [36]. The NQO2 enzymatic activity assay was performed in QR buffer with 90 µM of the co-substrate 1-benzyl-1,4-dihydro-nicotinamide (BNAH) from Santa Cruz Biotech., (208609), and 90 µM of substrate, menadione (Supelco, Merk Life Science S.r.l., Milan, Italy), in a total volume of 180 µL, incubated at 25 °C. After the addition of 10 ng/mL of recombinant human (h)NQO2 or 5 μg of protein extracts, the reaction was initiated and monitored within 10–20s at fluorimeter for (96-well plate reader, 1420 Multilabel reader Victor 2, Perkin Elmer). Enzyme kinetics of the oxidation of BNAH was followed at 440 nm with excitation at 340 nm. The slope of the decrease of BNAH fluorescence was analyzed by Microsoft Excel. The reaction velocity (V) was determined by fitting the linear regression for the first 6 to 8 min of the reaction and calculating the absorbance decrease in the first 5 min of the reaction, which was then normalized to BNAH concentration and expressed in micromole per minute (μM × min^−1^). Then, this value was used to calculate the enzyme activity (EA) expressed in μmol min^−1^ per mg of protein or enzyme source (μmol min^−1^ mg^−1^) and corresponded to the maximal activity of the oxidoreduction reaction in these conditions. For inhibition measurements, flavonoids were dissolved in ethanol, as stock solution (15 mM), and appropriate dilutions were performed in assay buffers. For the calculation of IC_50_ values, 7 to 8 dilutions of inhibitors were assessed, and EA were calculated for each reaction. Then, data were analyzed by GraphPad PRISM 8.0 (GraphPad Software, San Diego, CA, USA) for IC_50_ for each flavonoid.

### 2.6. Protein Extraction and Western Blot Analysis

At indicated times after the treatments, cell monolayers were washed once with PBS and scraped in cold Lysis buffer 1 (LB1) containing 50 mM Tris-HCl, pH 7.3, 150 mM NaCl and the following reagents from Sigma-Aldrich: 1% Igepal (I7771), 1 mM EDTA (E5134), phosphatase inhibitors (1mM NaF [S1504], 0.01 mM sodium orthovanadate [S6508], 10 mM β-glycerophosphate [G9422], and protease inhibitor cocktail (Mini tablets, Thermo Fisher Scientific, Waltham, MA, USA, 88665). The lysate was placed on ice for 5 or 10 min and then centrifuged at 4 °C, 15,000× *g* for 10 min. Then, the supernatant fraction was mixed (3:1) with Laemmli Sample buffer (4×: 100 mM Tris-HCl, pH 6.8, 4% SDS (Sigma-Aldrich, L4390), 16% glycerol, 20% 2-mercaptoethanol (Carlo Erba Reagents, 460691)), boiled for 15 min at 90 °C, and then stored at −80 °C until analysis. Samples were boiled again for 3 min at 90 °C before loading onto 12% or 8% polyacrylamide SDS (SDS-PAGE) gels. For analysis of serine-threonine phosphorylations (phAMPK), cell monolayers were lysed as described previously [39]. Protein concentration was quantified by BCA protein assay (Thermo Fisher Scientific, 23227). In general, 10–20 μg of protein lysate was loaded on 12% or 8% SDS-PAGE. After electrophoresis, polypeptides were transferred to PVDF membranes (Bio-Rad Laboratories Inc., Hercules, CA, USA(Bio-Rad), 162–0177), which were thereafter saturated with 5% milk (Panreac, AppliChem GmbH, Darmstadt, Germany, A0830) in TTBS or PBS-T for 1 h at RT. The primary antibody was usually incubated overnight, followed by 1 h–incubation with a secondary antibody. Blots were developed with the enhanced chemiluminescence (ECL) procedure, using ECL reagents (ImmunoStar^®^, Bio-Rad, 170–5070), digitally acquired using a ChemiDoc XRS imaging system (Bio-Rad). The full-size WB blot images for the data presented in Figures 3–6 are shown in Appendix A.

### 2.7. Statistical Analysis

The results were expressed as the mean ± standard error (SEM). WB optical density (OD) was analyzed as previously described by Quantity One^®^ 1-D analysis software (Bio-Rad) [36], and normalized to the mean OD of the first three bands in each blot and then to the mean of control bands in a set of analyzed blots from one experiment. The data were evaluated with GraphPad Prism 8.0.0 using one-way ANOVA followed by Dunnett’s or Bonferroni multiple comparisons tests, as suggested by the software, or Brown–Forsythe and Welch ANOVA followed by Tamhane’s T2 multiple comparisons test as appropriate. When not indicated, the T test was used for statistical analysis. The between-group differences were considered significant at *p* < 0.05.

## 3. Results

We tested if there was a correlation between the NQO2 inhibitor properties of flavonoids and their ability to induce autophagy. We focused on six flavonoid aglycons, found in fruits and leaves of Citrus family plants such as bergamot [15], which are also the main compounds of Bergamot Polyphenol Fraction (BPF)^®^, previously characterized for its pro-autophagic effects in hepatocytes [7,38]. Here, BPF flavonoids were compared to quercetin, a flavonol, which is known to induce autophagy in different systems [40,41]. This was done in human hepatoma HepG2 cells stably expressing GFP-LC3-DsRed, a double fluorochrome LC3 protein (GR-LC3-HepG2). The latter cell line was developed to quantify LC3-II induction easily by an alternative and more sensitive method than Western blotting (WB) [7]. In this assay, the ratio between red and green fluorescence (autophagy index, ATG index), measured by flow cytometry indicates the amount of lipidated LC3 (LC3-II, red) with respect to unprocessed LC3 (red and green), which correlates with an increase in autophagosome formation (Figure 1A). ATG index measurement was performed on cells cultured in high-glucose medium and exposed for 24 h to 0.3 mM PA, to induce hepatic steatosis-like state in vitro, since in these conditions, flavonoids have been shown to stimulate a stronger autophagic response [38]. Treatment of GR-LC3-HepG2 with a low concentration of flavonoids (5 μg/mL ≈15 μM) led to a marked increase of ATG index in the cases of apigenin, luteolin, and quercetin and a tendency to enhance autophagy in case of diosmetin (Figure 1B).

NQO2 activity assays were performed on HepG2 cell protein extracts, using its specific co-substrate BNAH, therefore avoiding the interference with NQO1. NQO2 activity was strongly inhibited by luteolin, apigenin, moderately by quercetin, and weakly by diosmetin, while marginally by the other three polyphenols at the lower (5 μM) concentration tested (Figure 2A). Among seven tested flavonoids, apigenin and luteolin followed by quercetin were the strongest inhibitors of NQO2 activity when added to the pure human recombinant NQO2, demonstrating that the inhibitory effect of flavones is direct (Figure 2B,C). The calculated IC50 values in this assay were below the micromolar range for luteolin, apigenin, and quercetin and ten to one hundred times higher for other flavonoids (Figure 2D,E). The present data demonstrate that the inhibitory activity of Citrus polyphenols on NQO2 correlates with their pro-autophagic activity, suggesting that NQO2 may be involved in autophagy regulation.

To address this hypothesis, we used a transient transfection of specific siRNA to silence NQO2 expression. HepG2 cells transfected with a control siRNA and NQO2-specific siRNA were treated with luteolin and apigenin (10 μg/mL) for 6 h and analyzed by WB for LC3-II modulation (Figure 3A).

The silencing approach based on a mixture of different anti-NQO2 siRNA limited the expression of NQO2 by 70% to 80%, which was much more than previously achieved [32,36,42]. The treatment with both flavones caused a significant upregulation LC3-II within 6 h in control cells expressing a control siRNA but had no effect on total p62 levels at this time point. This occurred both in the presence or absence of chloroquine (ClQ), indicating a stimulation of the autophagic flux and NQO2 expression by these flavones. In contrast, cells with reduced expression of NQO2 by siNQO2 did not respond to flavones with a significant increase of LC3-II levels regardless the co-treatment with ClQ (Figure 3A). These observations were reproducibly obtained in four independent experiments and further analyses of the data by densitometry revealed that the silencing of NQO2 slightly but significantly augmented the basal autophagic flux in HepG2 cells, beside preventing further LC3-II induction in response to flavones (Figure 3B). In addition, we observed a reproducible induction of NQO2 expression by luteolin and apigenin (Figure 3C), which might reflect an ARE-mediated transcriptional stimulation of NQO2 promoter by NRF2, which is a well-known effect induced by flavones on ARE-regulated phase II and antioxidant enzymes [43,44].

Next, we addressed the question whether this was related to the catalytic activity of the enzyme. We reasoned that if flavones, which inhibit NQO2, can induce autophagy, the molecules working as NQO2 substrates that stimulate its activity should inhibit autophagy. Menadione (K3) is a quinone vitamin K derivative and a substrate of NQO2. The addition of K3 activates the cellular NQO2 activity even in the absence of an exogenously added co-substrate [36]. Indeed, HepG2 cells exposed to K3 showed lower autophagic flux than control cells at 2, 6, and 20 h post-treatment, since LC3-II was reduced compared to vehicle-treated cells in both the presence and absence of ClQ (Figure 4A). However, another NQO2 substrate, adrenochrome, did not cause any significant downregulation of autophagic flux (Figure 4B), even in the presence of NQO2-specific co-substrate BNAH (Figure 4C) at early times post-treatment, although a weak inhibitory effect was observed after 20 h in both conditions.

Next, we addressed what can be the downstream signaling pathways stimulated by flavone binding and inhibition of NQO2. A likely candidate is AMPK. In addition to its role as a central sensor of cellular bioenergetic status [22], it is also an established inducer of autophagy and several publications have demonstrated a role of AMPK in polyphenol-induced autophagy [8,41,45]. First, we tested if luteolin and apigenin stimulate AMPK activity in HepG2 cells in our culture conditions. Different concentrations of flavones were applied for 6 h in HepG2 cells, and AMPK activation was analyzed with anti-phAMPK antibodies by WB. We observed a significant increase in T183/172 AMPK phosphorylation with 10 µg/mL luteolin or apigenin but not with higher concentrations of flavones (30 µg/mL) (Figure 5A). Next, to demonstrate that this AMPK activation is necessary for autophagy stimulation in HepG2 cells, we suppressed AMPK induction by silencing the beta1 subunit of AMPK (AMPKβ1) complex known as PRKAB1 [46]. Silencing of PRKAB1 in apigenin-treated HepG2 cells strongly reduced AMPKα phosphorylation (upper panel) and autophagic flux in response to apigenin (Figure 5B,C). Similar data were obtained for luteolin, even though it was a weaker inducer of AMPK phosphorylation compared to apigenin (Figure 5B,C). Indeed, LC3-II levels were elevated after PRKAB1 silencing in flavone-untreated cells, but they were not further increased by flavones nor by ClQ (Figure 5C), suggesting that LC3-II accumulation occurred due to a reduced autophagy flux in cells with suppressed AMPK activity. These data indicate that AMPK activation is necessary for flavone-mediated autophagy stimulation in HepG2 cells.

Lastly, we addressed the relationship between AMPK activation and levels of NQO2. To this end, NQO2 expression was suppressed by siRNA. HepG2 cells were treated with luteolin and apigenin (as in the experiments described in Figure 3A), lysed, and then analyzed by WB. Anti-NQO2 siRNA suppressed the levels of NQO2 by around 70%, which caused a marked increase in basic AMPK phosphorylation in control vehicle-treated cells. Importantly, there was no further increase of phAMPK signal in the response to flavones (Figure 6A), which was confirmed by statistical analysis of OD in 4 independent experiments (Figure 6B). In addition, in cells expressing low levels of NQO2, luteolin showed no statistically significant tendency to suppress the basic AMPK phosphorylation (Figure 6A,B). Thus, the silencing of NQO2 increases the steady-state AMPK activation and inhibits the response to flavones in HepG2 cells, suggesting an antagonism between activable NQO2 and AMPK.

## 4. Discussion

The enzymes catalyzing specific redox reactions in cells not only contribute to oxidative balance and stress regulation, but they often play other critical functions in the cell. The data presented in this paper support such a scenario for NQO2 and indicate that NQO2, beside a role in quinone metabolism, regulates autophagy. The most important finding is the demonstration that the downregulation of NQO2 levels by siRNA prevents efficient induction of LC3-II by flavones, suggesting a role of this enzyme in the regulation of autophagy machinery. In particular, the decrease of the NQO2 protein levels by “siRNA-mediated knock-down” increased the basal LC3-II levels, but it inhibited its further up-regulation by luteolin and apigenin (Figure 3A,B). This mechanism may depend on the inhibition of the catalytic activity of NQO2, since the pro-autophagic activity of flavonoids correlated very well with their ability to inhibit the enzyme activity. Therefore, it would be conceivable that the products of NQO2 activity, such as hydroquinones or ROS, generated during this process might inhibit some steps in the autophagy process, such as in the case of menadione (Figure 4).

However, the effects of adrenochrome plus BNAH do not support this hypothesis. This substrate/co-substrate pair has little effect on LC3-II levels, although we cannot exclude that adrenochrome stimulates other pro-autophagic pathways independently of NQO2, leading to a neutralization of the inhibitory effect of NQO2. We cannot exclude that these are distinct phenomena and that the generation of hydroquinones or related metabolites that naturally accompany NQO2 activity has little to do with autophagy modulation. The question of whether active or rather activable (ligand-free) NQO2 suppress autophagy could be addressed by site-specific mutagenesis of NQO2 and generation of enzyme-dead and constitutively active mutants, but so far, this objective has failed, as none of the mutants we tested was totally inactive or insensitive to ligands [47]. In addition, the activity of NQO2 is difficult to establish in living cells, because it depends not only on the levels of quinones but also on the availability of its co-substrates such as nicotinamide mononucleotide (NMN) or related compounds [48]. That co-substrate availability remains unexplored, and it is unknown how they are generated in the cell [26]. In fact, the NQO2 activity assay is based on substrates and co-substrates provided externally and does not measure the actual cellular activity of NQO2 but rather its potential activity. Therefore, it cannot be excluded that the binding of flavones to NQO2 might generate a conformational change in the three-dimensional structure of the protein, as reported for its ligand chloroquine [49] and may impact on its interaction with other proteins. As a matter of fact, NQO2 has been reported to mediate some of its function via protein–protein interactions. The most important NQO2 partners might be the tumor suppressor p53, myeloid differentiation factor CCAAT-enhancer-binding protein α (C/EBPα) [33], or the oncogene AKT-1 [50]. For example, a functional crosstalk between AKT-1 and NQO2 is likely mediated by a physical interaction involving the chains A and B in the vicinity of resveratrol-binding domain of NQO2. The authors proposed that this interaction mediates the pharmacological effects of resveratrol in tumor cells [50]. It is established that the suppression of AKT-1 activity inhibits mammalian target of rapamycin (mTOR) complex 1, which negatively regulates autophagy and AMPK activity. However, the findings regarding the interaction of NQO2 with Akt-1 have to be consolidated before drawing any conclusions related to the actual mechanism of autophagy regulation by NQO2 and flavones.

It is likely that besides these few known interaction partners of NQO2, other proteins, such as regulators of autophagy, might interact upon the binding of NQO2 to flavones or other ligands, such as S29434/NMDPEF. For sure, beside a potential role in autophagy pathway, the data presented in the present work reveal a direct role of NQO2 as a novel polyphenol sensor in cells, modulating cell metabolism and mediating the biological response to natural polyphenols. Thus, NQO2 can act in parallel to PDE that degrades cyclic AMP necessary to trigger a release of Ca^+2^ from intracellular calcium stores and, upon few other transduction steps, to the activation of AMPK [24] (Figure 7).

For this reason, we postulated that AMPK activation might be also downstream of NQO2 as it is downstream of PDE. This hypothesis is supported by many other observations. For example, AMPK is one of the most important targets of resveratrol and other polyphenols in different cell systems [8,24,51]. AMPK, in addition to its role as a central player of fat and carbohydrate metabolism [22,52], is also an established inducer of autophagy. Indeed, it directly phosphorylates mammalian-initiating kinase Ulk1 and suppresses mTOR complex 1 (TORC1) activity, which is the triggering event for autophagy in mammalian cells [53,54] (Figure 7). Moreover, the silencing of AMPK has been shown to reduce epigallocatechin-3-gallate, which is a flavonoid derivative that induced autophagy in HepG2 cells [8]. Using a similar approach, we managed to confirm that both luteolin and apigenin determined the activation of AMPK and that it is necessary for autophagy induction in HepG2 cells, which could be deduced from related findings in the literature [8,55,56,57]. The most important evidence that AMPK activation depends on NQO2 status (flavone-free versus flavone-bound) is the fact that the silencing of NQO2 causes an increase in steady-state AMPK phosphorylation, suggesting that NQO2 might directly or indirectly inhibit AMPK activation. Interestingly, when NQO2 is absent (or very low), apigenin and luteolin are unable to cause a further increase in AMPK phosphorylation. This suggests that the presence of the NQO2–flavone complex is necessary for the modulation of AMPK activity. In the absence of NQO2, the activity of AMPK is high and insensitive to polyphenol-dependent regulation at least in Hep2 cells. At present, we are not able to explain how NQO2 suppresses AMPK activity, and future studies should address this interesting mechanism.

## 5. Conclusions

The data presented here show for the first time that targeting NQO2 by flavones stimulates pro-autophagic signals and thus confirms a regulatory role of this multitask-enzyme in autophagy. Our findings have important implications not only for autophagy but for all AMPK-regulated functions, including the suppression of liver lipogenesis and glycogen synthesis, induction of fatty acid oxidation and glucose uptake, and other effects that protect from NAFLD and diabetes [22,58]. Although the exact mechanism of the crosstalk between NQO2 and AMPK and/or other autophagy regulators remains unexplored, it is a very encouraging step forward in our understanding of molecular pathways and biological processes involving NQO2.

## Figures and Tables

**Figure 1 antioxidants-10-00776-f001:**
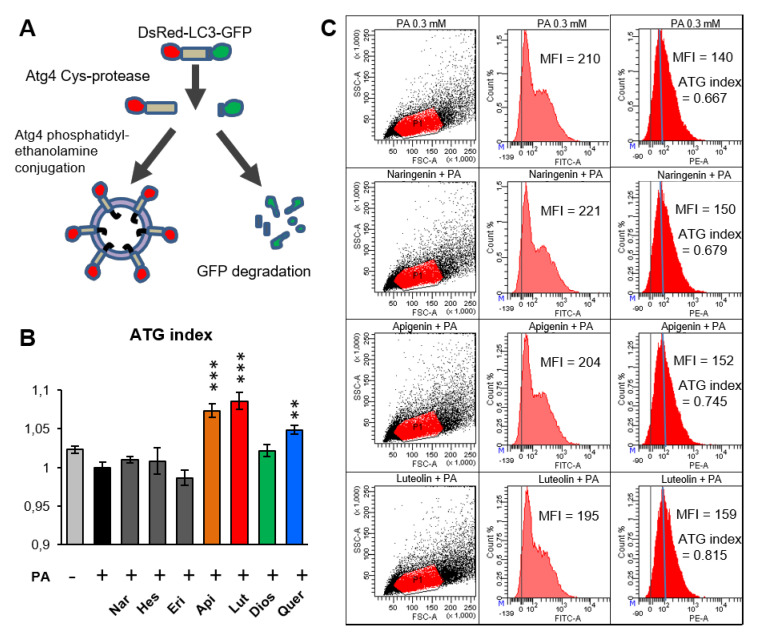
Comparison of pro-autophagic activity of seven different flavonoid aglycones reveals that flavones and quercetin are the strongest inducers of autophagy. (**A**) Description of the system used to quantify autophagy. Double-fluorescent DsRed-LC3-GFP is stably expressed in hepatoma HepG2 cells, giving rise to GR-LC3-HepG2 cells. When LC3 is being processed into LC3-II the C-terminal GFP is removed, but LC3-II remains linked to N-terminal DsRed fluorochrome. Autophagy (ATG) index is calculated as the ratio between red (LC3-II) and green fluorescence intensity (unprocessed LC3- GFP-linked). (**B**) GR-LC3-HepG2 cells were pre-treated with palmitic acid (PA) for 24 h and then treated for 6 h with one of the following aglyconic flavonoids (5 µg/mL): naringenin (Nar), hesperetin (Hes), and eriodictyol (Eri), diosmetin (Dios), apigenin (Api), luteolin (Lut), and quercetin (Quer). After the trypsin-mediated detachment of cell monolayers, the ATG index was analyzed by flow cytometry. The graph shows the ATG index normalized to PA-treated control cells (mean ± SEM) in a representative experiment performed in triplicate. Statistical analysis: one-way ANOVA followed by Dunnett’s test; ** *p* < 0.01, *** *p* < 0.001; significantly different from PA-treated control (black bar). (**C**) Scatter plots and profiles of green (FITC-A) and red (PE-A) fluorescence of four representative samples from the experiment presented in (**B**). Mean fluorescence intensity (MFI) values for red and green channels and resulting ATG indexes calculated for the presented plots are shown.

**Figure 2 antioxidants-10-00776-f002:**
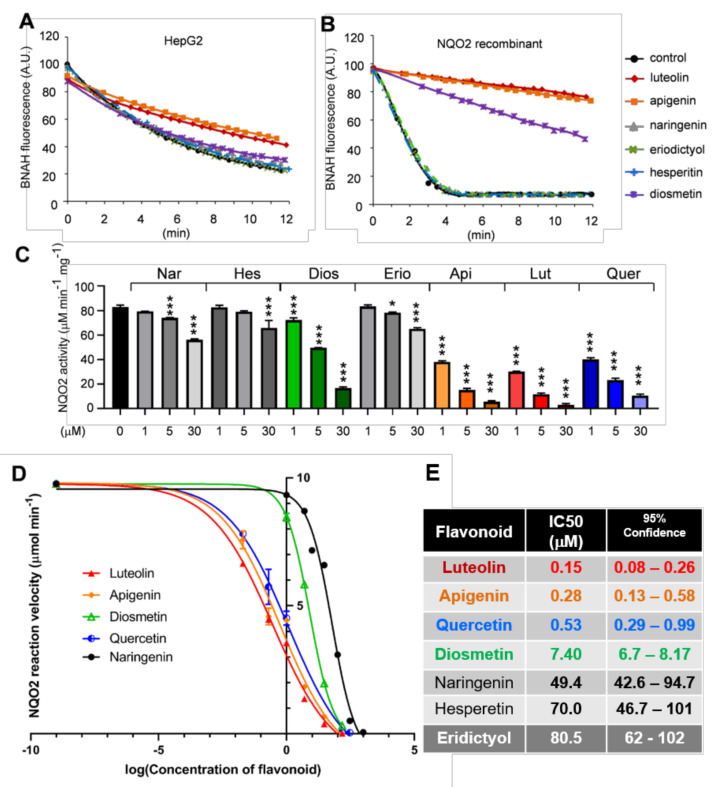
Inhibitory activity of *Citrus* polyphenols on NQO2 correlates with their pro-autophagic activity. Fluorimetric determination of NQO2 co-substrate (BNAH) oxidation in the presence of different sources of NQO2: (**A**) HepG2 cells (5 µg/well) and (**B**) purified human recombinant NQO2 (hNQO2, 10 ng per well) and 5 μM of flavonoids. Experimental points and best-fitted curves to are shown. (**C**) Flavones and flavonol quercetin are the best inhibitors of NQO2 activity. The reactions were carried out as described above in the presence of polyphenols at indicated concentrations and NQO2 activity was calculated. The graph shows the mean ± SEM of an experiment performed in triplicate. Statistical analysis: one-way ANOVA followed by Dunnett test; * *p* < 0.05, *** *p* < 0.001; significantly different from control. (**D**) The enzyme inhibition curves for hNQO2 used to calculate IC_50_ values. The curves for five inhibitors are shown. (**E**) IC50 values calculated for the assays as in C and D.

**Figure 3 antioxidants-10-00776-f003:**
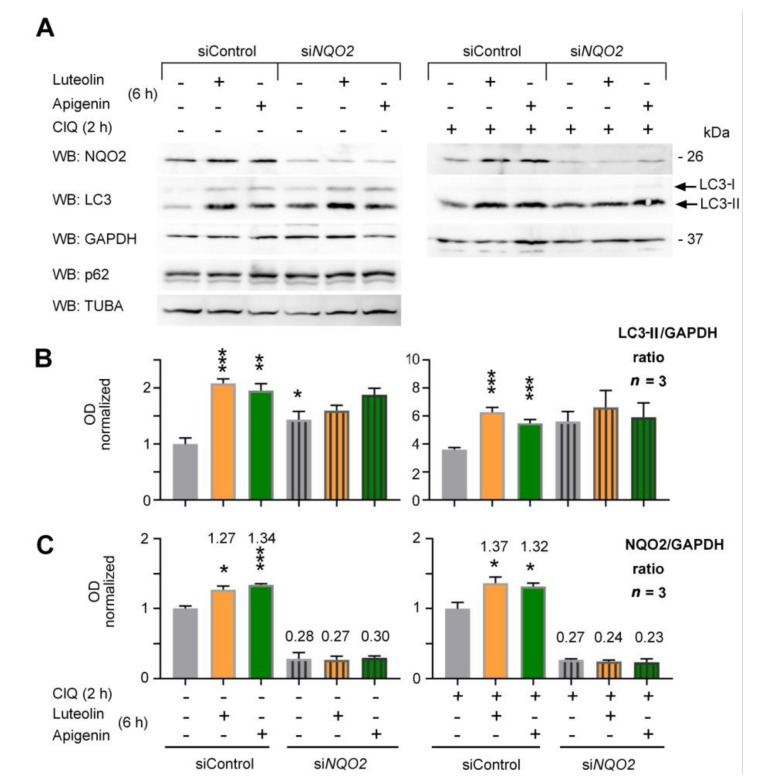
Silencing of NQO2 augments autophagic flux in HepG2 cells and reduces the response to luteolin and apigenin. (**A**) HepG2 cells were transfected with a control siRNA and NQO2-specific siRNA, which were treated as indicated for 6 h (luteolin, apigenin 10 g/mL) and analyzed by WB (SDS-PAGE 12%) for the expression of NQO2 and LC3. GAPDH was used as loading control. (**B**,**C**) The graphs show the OD analysis of LC3-II (**B**) and NQO2 (**C**) (ratio to GAPDH, normalized to vehicle-treated controls) of the experiment described in (**A**) as means +/− SEM of three blots from *n* = 3 independent experiments, each analyzed at two different exposures. Statistical analysis: Brown–Forsythe and Welch ANOVA followed by Tamhane’s T2 test; * *p* < 0.05, ** *p* < 0.01, *** *p* < 0.0001.

**Figure 4 antioxidants-10-00776-f004:**
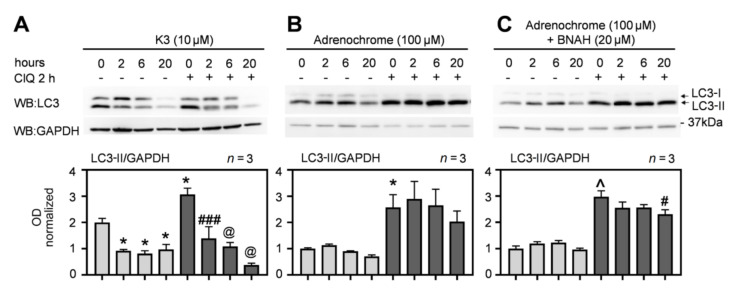
Suppression of basal autophagy by stimulation of NQO2 activity depends on its substrate. (**A**) HepG2 cells were exposed to menadione (K3), a known NQO2 substrate and treated or untreated with 25 μM ClQ for 2 h before lysis, lysed at the same time and LC3-II was evaluated by WB (SDS-PAGE 12%). (**B**,**C**) Cells were treated with 100 μM adrenochrome -/+ BNAH (20 μM) for indicated times to stimulate cellular NQO2, then with 50 μM ClQ 2 h before lysis and analyzed as above in A. The graphs below show the results of OD analysis from three independent experiments. Statistical analysis: one-way ANOVA followed by Bonferroni’s post-test. * *p* < 0.05, ^∧^ *p* < 0.0001 when compared to vehicle (0, -) or ^#^ *p* < 0.05, ^###^
*p* < 0.001, ^@^ *p* < 0.0001 when compared to ClQ (0, +) control samples.

**Figure 5 antioxidants-10-00776-f005:**
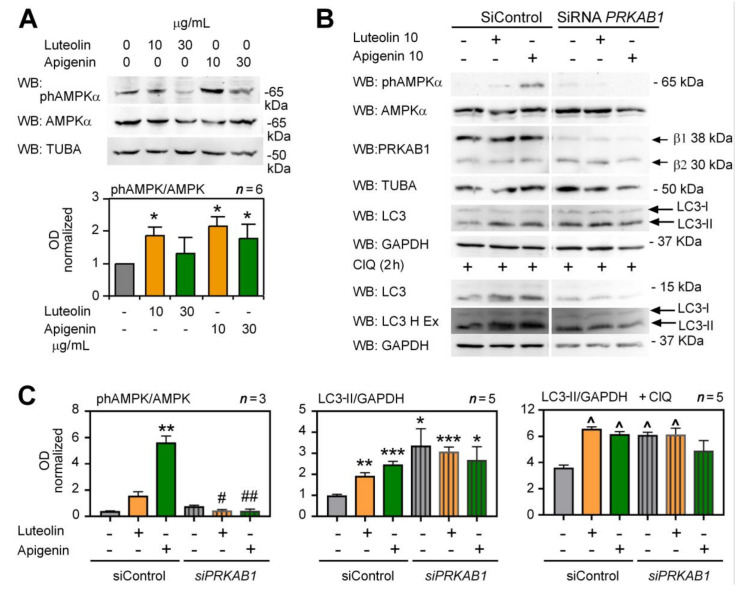
AMPK activation is necessary for autophagy induction by apigenin and luteolin in HepG2 cells. (**A**) Both apigenin and luteolin stimulate AMPK phosphorylation in dose-dependent fashion. HepG2 cells were treated with 10 or 30 μg/mL of luteolin and apigenin for 6h and phAMPK levels (phT183/172 in AMPK a1/2) were analyzed by WB (SDS-PAGE 8%). AMPK was used as a loading control. The graph below shows the densitometric analysis of data from three independent experiments reloaded two times each (*n* = 6). (**B**) Suppression of AMPK activation by silencing of AMPK beta1 subunit (PRKAB1) reduces autophagic flux and response to apigenin and luteolin. HepG2 cells were transfected with a control siRNA or *PRKAB1*-specific siRNA, treated for 6 h, and analyzed by WB for the expression of indicated above proteins. ClQ (50 μM) was added 2 h before lysis to enhance LC3-II accumulation. (**C**) OD analysis of data in (**B**) from *n* = 3 independent experiments, each analyzed twice at two different exposures. LC3-II in the presence of ClQ was normalized to its mean induction by ClQ (2 h, 50 μM) when compared to control (-, -) samples. Statistical analysis: Brown–Forsythe and Welch ANOVA followed by Tamhane’s T2 test; * *p* < 0.05, ** *p* < 0.01, *** *p* < 0.001, ^∧^ *p* < 0.0001 when compared to control samples or ^#^ *p* < 0.05, ^##^ *p* < 0.01 when compared to *siPRKAB1-*treated control samples.

**Figure 6 antioxidants-10-00776-f006:**
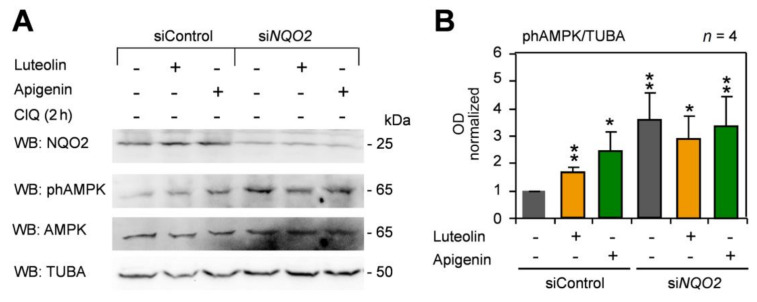
Silencing of NQO2 augments basic AMPK phosphorylation and inhibits the response to apigenin and luteolin. (**A**) HepG2 cells were transfected with a control siRNA and NQO2-specific siRNA, treated with flavones (10 μg/mL) for 6h and analyzed by WB for phosphor-AMPK, total AMPK, LC3-II, and for the levels of NQO2 to verify the efficacy of anti-*NQO2* siRNA. TUBA expression was analyzed as a loading control. (**B**) OD analysis (ratio to TUBA, normalized to vehicle-treated controls) of phAMPK signal as means +/− SEM of n blots from *n* independent experiments. Statistical analysis: student T-test; * *p* < 0.05, ** *p* < 0.01.

**Figure 7 antioxidants-10-00776-f007:**
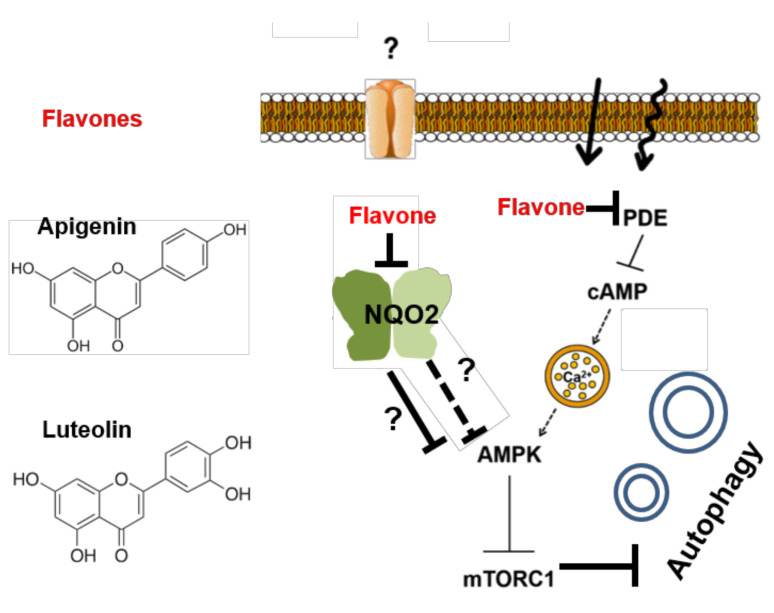
Role of NQO2 in flavonoid-induced autophagy. Flavones or other flavonoids enter the cytoplasm by means of specific receptors or free diffusion and bind to NQO2 or other receptors such as PDE. Ligand-free (activable) NQO2 prevents AMPK activation by unknown direct or indirect mechanisms, while the binding of a flavonoid allows AMPK phosphorylation, leading to mTORC1 inhibition and the stimulation of autophagy. This process is likely independent of NQO2 activity, but it positively correlates with NQO2 inhibition by flavonoids. NQO2 and PDEs may synergize or have an additive effect on AMPK activation and its downstream consequences including autophagy.

## Data Availability

The data is contained within the article and Appendix A.

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
