# Peer review of "Apigenin and Luteolin Regulate Autophagy by Targeting NRH-Quinone Oxidoreductase 2 in Liver Cells"

_antioxidants, 2021, doi:10.3390/antiox10050776_

Round 1
Reviewer 1 Report
The paper entitled “Apigenin and luteolin regulate autophagy by targeting NRH- 2 quinone oxidoreductase 2 in liver cells.” presented by Janda E et al. for publication in the Journal Antioxidants is part of articles concerning the analysis of dietary flavonoids and their influences on autophagy fluxes and the related signaling pathways.
The experiments are well conducted and the results are convincing. However, I have several major concerns that need to be answered before this work could be considered to be published:
Major remarks
Figure 2B: A biparametric histogram should be added to this figure, as a proof of concept: an representative image between treated and untreated cells. The description of flow cytometry and the corresponding settings (lasers, filters, voltage,…) are missing in the material and methods section. The Atg index is not described neither.
In the legend of figure 2B, the description of the different treatments is not clear. Does it mean that cells were pre-treated with PA, then one of the flavonoids (Nar, Hes, Eri), before being treated with the last flavonoids (Dios, Api, Lut, Quer)? Then results are the combination of 2 flavonoids?
Figure 3C is described line 261, but is absent from figure 3.
Figure 4: These experiments should be repeated and quantification added to the figure.
Figure 5A: the loading control (tubulin) should be presented.
Figure 5B and C: the immunoblots with anti-LC3 antibodies reveals only one band. It should be repeated and 2 bands LC3-I and II should be presented.
Silencing PRKAB1 reduces phosphorylation of AMPKα. However, total AMPKα is not shown.
It is obvious from this figure that PRKAB1 knock down is also inhibiting AMPKβ1 and 2. This is not described in the manuscript. All quantifications of these blots should be presented, as mentioned in the legend.
Figure 6: total AMPK immunoblots should be presented.
Minor remarks
Line 209: “a weaker although statistically significant increase of autophagy in case of diosmetin (Fig. 1B).” There is no sign of significativity on the graph.
Line 223: To NQO2 activity assays were performed … Misspelling.
Line 346: “In addition, in cells expressing low levels of NQO2 luteolin tended to suppress the basic AMPK phosphorylation (Fig. 6A), which was confirmed by statistical analysis of OD in 4 independent experiments (Fig. 6B).” In this figure, there is no A and B labelling.
Line 418: “Ca+2” instead of Ca2+
Author Response
Comments and Suggestions for Authors
The paper entitled “Apigenin and luteolin regulate autophagy by targeting NRH- 2 quinone oxidoreductase 2 in liver cells.” presented by Janda E et al. for publication in the Journal Antioxidants is part of articles concerning the analysis of dietary flavonoids and their influences on autophagy fluxes and the related signaling pathways.
The experiments are well conducted and the results are convincing. However, I have several major concerns that need to be answered before this work could be considered to be published:
We thank the reviewer 1 for all valuable remarks regarding our paper and for considering it well conducted and convincing. We have performed all requested optical density (OD) analyses and implemented the paper with requested Western blots (WB), which have been also included as raw data in the supplementary material. All other requested data were provided and new Figures 1, 3, 4, 5 and 6 were prepared and included in the manuscript. The figure legends and the text of the manuscript have been corrected accordingly and the changes are in red. We hope that the reviewers appreciate substantial improvements in the figures and recognize the quality of the revised manuscript.
Major remarks
Figure 2B: A biparametric histogram should be added to this figure, as a proof of concept: an representative image between treated and untreated cells. The description of flow cytometry and the corresponding settings (lasers, filters, voltage,…) are missing in the material and methods section. The Atg index is not described neither.
Considering the remarks, it is obvious that the reviewers commented the Figure 1B, not 2B and for this reason our answer refers to the data presented in Fig. 1B.
We thank the reviewer for this valuable criticism. In fact, the description of the method was mistakenly omitted in the Material and methods section. As requested, we have now provided all the missing information including the details of fluorescence acquisition and ATG index calculation and four representative bi-parametric histograms for the data in Fig. 1B (see Fig. 1C).
In the legend of figure 2B, the description of the different treatments is not clear. Does it mean that cells were pre-treated with PA, then one of the flavonoids (Nar, Hes, Eri), before being treated with the last flavonoids (Dios, Api, Lut, Quer)? Then results are the combination of 2 flavonoids?
Indeed, the description was not clear. I hope the new description does not leave much room for doubt. “B) GR-LC3-HepG2 cells were pre-treated with palmitic acid (PA, 0.3 mM) for 24 h and then treated for 6 h with one of the following aglyconic flavonoids (5µg/ml): naringenin (Nar), hesperetin (Hes) and eriodictyol (Eri), followed by diosmetin (Dios), apigenin (Api), luteolin (Lut) and quercetin (Quer). After processing of the cell monolayer ATG index was analyzed by flow cytometry.”
Figure 3C is described line 261, but is absent from figure 3.
The Fig. 3C refers to the lower graphs reporting the OD analysis results for NQO2 expression. The letter C has been now added to the Fig. 3.
Figure 4: These experiments should be repeated and quantification added to the figure.
All the experiment presented in this manuscript have been performed at least 3 times. The relative OD quantifications have been now performed and a new version of the Fig. 4 now shows 3 new graphs, as requested. In addition, we exchanged the previous WB panels with the same data at higher resolution. The reference to the statistical analysis applied has been also added to the legend of Fig.4.
Figure 5A: the loading control (tubulin) should be presented.
The tubulin blot has been added to the Fig. 5A and to the supplementary material, as requested.
Figure 5B and C: the immunoblots with anti-LC3 antibodies reveals only one band. It should be repeated and 2 bands LC3-I and II should be presented.
Since the data in the figure 5B and 5C show the alternative data from the same type of PRKAB1 silencing experiment, we removed the old figure 5B and implemented the Fig.5C (now Fig.5B) with missing data such as total AMPKalpha, beta (PRKAB) and LC3 +/- chloroquine (ClQ). Furthermore, as requested the entire data set shown in the new Fig.5B was analyzed by densitometry, as requested.
Unfortunately, we often experienced with HepG2 cells that the upper LC3-I band was scarcely visible at short exposures with the antibody and blotting conditions that we used, especially in the presence of ClQ. However, at longer exposures the LC3-I band could be detected in most of the cases. To solve the problem pointed out by the reviewer, a longer exposure blot was presented for the Fig.5B (old figure C). We hope that the reviewer agrees that this solution should be sufficient, rather that repeating the entire experiment and trying to avoid this technical problem.
Silencing PRKAB1 reduces phosphorylation of AMPKα. However, total AMPKα is not shown.
It is shown now in Fig. 5B.
It is obvious from this figure that PRKAB1 knock down is also inhibiting AMPKβ1 and 2. This is not described in the manuscript. All quantifications of these blots should be presented, as mentioned in the legend.
PRKAB is just an updated name of AMPKβ. The comparison of data from other independent experiments did not reveal any significative downregulation of AMPKβ2 (PRKAB2, lower band) in our system. In fact, AMPKβ2 seemed to be clearly downregulated only in 1 sample and we have removed the set of data containing this sample. Now, AMPKβ2 blot shown in the new Fig. 5B (Old Fig. 5C) is more representative and there is no downregulation of AMPKβ2.
Figure 6: total AMPK immunoblots should be presented.
The blot for total AMPK has been added to the Fig. 6 and to the relative original blot has been presented in the raw supplementary material. Nevertheless, the quantification of phospho-AMPKa can be well done compared to a loading control such as tubulin or another house -keeping gene, since the levels of total AMPK can vary under certain conditions and sometimes total AMPKa does not reflect the loading control (Pineda et al, Cell 2015).
Minor remarks
Line 209: “a weaker although statistically significant increase of autophagy in case of diosmetin (Fig. 1B).” There is no sign of significativity on the graph.
According to our data the increase in diosmetin is significative only with student T test, but not with ANOVA plus Dunnett’s test. We are sorry for this mistake. The text in line 209 was changed to “…. led to a marked increase of ATG index in case of apigenin, luteolin and quercetin and a tendency to an increase in case of diosmetin (Fig. 1B)”
Line 223: To NQO2 activity assays were performed … Misspelling.
Thank you, corrected.
Line 346: “In addition, in cells expressing low levels of NQO2 luteolin tended to suppress the basic AMPK phosphorylation (Fig. 6A), which was confirmed by statistical analysis of OD in 4 independent experiments (Fig. 6B).” In this figure, there is no A and B labelling.
The letters A and B has been now added to the Fig. 6.
Line 418: “Ca+2” instead of Ca2+
Thank you, corrected.
Reviewer 2 Report
In Janda et al., the authors demonstrate that the proautophagic activity of apigenin and luteolin, two flavonoid aglycons, correlates with their capacity to inhibit NQO2 by AMPK activation in engineered HepG2 cells.
This manuscript is well written and structured and can be accepted for publication after the following comments being addressed:
- NQO2 is highly expressed in liver and kidney. Did the authors check the proautophagic activity of the flavonoids in not tumorigenic or in renal cell lines?
- The authors focused on the expression of LC3-II, did the authors checked the expressions of other markers in the autophagic cascade (Beclin 1, p62…)?
- The authors postulate the induction of NQO2 expression by luteolin and apigenin might reflect an ARE-mediated transcriptional stimulation of NQO2 promoter by NRF2 induced by flavones. To strenghtenes the results could be appropriate to check the NQO2 RNA levels in treated cells.
- The panel C is missing in the caption and in the Figure 3.
- In the same cell lines, the detection and the modulation of LC3 I is different, how the authors could explain that?
- In the Figure 4 are missing the quantification graphs.
Typos:
- Lines 143: 500 l of QR buffer as previously described…..
- Line 147: NQO2 or 5 g …..
- Line 298: We observed a significant increase in Th1234 AMPK
Author Response
Response to Reviewer 2
We thank the reviewer 2 for all valuable remarks regarding our paper and for considering it well written and well structured. We have addressed all the suggestions and comments pointed out by the reviewer 2. To this end we performed all requested optical density (OD) analyses and implemented the paper with requested Western blots (WB), which have been also included as raw data in the supplementary material. All other requested data were provided and new Figures 1, 3, 4, 5 and 6 were prepared and included in the manuscript. The figure legends and the text of the manuscript have been corrected accordingly and the text modifications are in red. We hope that the reviewers appreciate substantial improvements in the figures and recognize the quality of the revised manuscript.
Reviewer 2:
In Janda et al., the authors demonstrate that the proautophagic activity of apigenin and luteolin, two flavonoid aglycons, correlates with their capacity to inhibit NQO2 by AMPK activation in engineered HepG2 cells.
This manuscript is well written and structured and can be accepted for publication after the following comments being addressed:
- NQO2 is highly expressed in liver and kidney. Did the authors check the proautophagic activity of the flavonoids in not tumorigenic or in renal cell lines?
We have tested the proautophagic activity of flavonoids in other cell lines of hepatic, mammary epithelial and astrocytoma origins and in primary astrocytes, but not yet in renal cell lines. There is a correlation between NQO2 level and the autophagy induced by flavonoids, with some exceptions, but the data are not ready for publication yet.
- The authors focused on the expression of LC3-II, did the authors checked the expressions of other markers in the autophagic cascade (Beclin 1, p62…)?
p62 levels do not change at 6 h post treatment in HepG2 cells. We added a p62 blot to the figure 3. In the first version of the paper the Fig. 6 contained a p62 blot, which has been eliminated as it did not add any relevant information to this figure.
- The authors postulate the induction of NQO2 expression by luteolin and apigenin might reflect an ARE-mediated transcriptional stimulation of NQO2 promoter by NRF2 induced by flavones. To strenghtenes the results could be appropriate to check the NQO2 RNA levels in treated cells.
The time assigned for revision is too short to perform such an experiment. We cited papers that show a transcriptional modulation of NQO2 or its relative NQO1 by other antioxidants.
- The panel C is missing in the caption and in the Figure 3.
- The letter C refers to the lower graphs reporting the OD analysis results for NQO2 expression. The missing letter has been now added to the Fig. 3.
- In the same cell lines, the detection and the modulation of LC3 I is different, how the authors could explain that?
In cells treated with chloroquine to block the autophagic flux, we observe a decrease of LC3-I and an increase of LC3-II. Beside this obvious reason of LC3-I modulation, we also observe a slight induction of LC3-I in response to flavonoids and LC3-I changes in response to other factors that modulate autophagy. The strong increase of LC3-I, when autophagy is stimulated, is a well-known phenomenon in yeast, but not always observed in mammalian cells discussed widely in Guidelines for the use and interpretation of assays for monitoring autophagy (Klionsky et al. Autophagy 2012 and more recent editions). The slight differences in cell density as well as the quality of the primary polyclonal antibody solution used in WB between experiments may also influence the detection of LC3-I which is sometimes difficult to visualize in HepG2 cells.
- In the Figure 4 are missing the quantification graphs.
- The relative OD quantifications have been now performed and a new version of the Fig. 4
- now shows 3 new graphs, as requested.
Typos:
- Lines 143: 500 l of QR buffer as previously described…..
- Line 147: NQO2 or 5 g …..
- Line 298: We observed a significant increase in Th1234 AMPK
- Thank you, corrected.
Round 2
Reviewer 1 Report
First I would like to thank the authors for their reply.
I still have few minor remarks:
- in the legend of figure 1b: "and then treated for 6 h with one 211 of of the following aglyconic flavonoids"
- If I am not mistaken, Tubulin blot (Fig 5A) is not presented in the sup data
Author Response
We would like to thank the reviewer 1 for his/her time and effort to review our manuscript.
According to the second review R2, the following corrections have been introduced:
- The legend of figure 1b has been corrected
- The tubulin-alpha (TUBA) blots have been added to the supplementary data file